# Flaws can be Applause: Unleashing Potential of Segmenting Ambiguous Objects in SAM

**Chenxin Li[1]**[*], **Yuzhi Huang[2]**[*], **Wuyang Li[1]**, **Hengyu Liu[1]**,
**Xinyu Liu[1]**, **Qing Xu[3]**, **Zhen Chen[4]**, **Yue Huang[2]**, **Yixuan Yuan[1]**[†]
[1]The Chinese University of Hong Kong [2]Xiamen University
[3]University of Nottingham Ningbo China [4]Yale University

## Abstract

As the vision foundation models like the Segment Anything Model (SAM) demonstrate potent universality, they also present challenges in giving ambiguous and uncertain predictions. Significant variations in the model output and granularity can occur with simply subtle changes in the prompt, contradicting the consensus requirement for the robustness of a model. While some established works have been dedicated to stabilizing and fortifying the prediction of SAM, this paper takes a unique path to explore how this flaw can be inverted into an advantage when modeling inherently ambiguous data distributions. We introduce an optimization framework based on a conditional variational autoencoder, which jointly models the prompt and the granularity of the object with a latent probability distribution. This approach enables the model to adaptively perceive and represent the real ambiguous label distribution, taming SAM to produce a series of diverse, convincing, and reasonable segmentation outputs controllably. Extensive experiments on several practical deployment scenarios involving ambiguity demonstrates the exceptional performance of our framework. Project page: `https://a-sa-m.github.io/`.

## 1 Introduction

The advent of Visual Foundational Models (VFMs) such as the Segment Anything Model (SAM) [22] has been unprecedented, largely due to the availability of vast datasets and computational resources. These models have exhibited remarkable generalization capabilities in zero-shot scenarios and the capacity to interact with human feedback. SAM, in particular, employs a specialized data engine to manage 11 million image masks, using a unique prompt-based segmentation framework to generate accurate masks for any object within a visual context, largely extending the capacity and generality of segmentation [27]. Such successes have been widely extended to various domains, such as medical imaging analysis [8, 35, 28], remote sensing [48], etc.

However, it has been observed that SAM suffers from severe *predictive ambiguity* to segment desired concepts [22] in practical scenarios. To uncover the factual basis, we delve into this ambiguity and detail it into two flaws according to experimental insight. Specifically, the first flaw lies in that SAM prediction is sensitive to slightly different prompt variants. As shown in Fig. 1 (a), we ground SAM into a realistic clinical scenario to segment the lesion in CT images. Even though three medical experts uniformly give rational box prompts covering the lesion, SAM, unfortunately, makes enormous differences among them, even including one wrong case. Further, we provide detailed statistics regarding IoU over the small perturbation (only 5 pixels) of box prompts, as shown in Fig. 1

---

[*]Equal Contribution: {chenxinli@link.cuhk.edu.hk, yzhuang13@stu.xmu.edu.cn}.
[†]Corresponding Author: {yxyuan@ee.cuhk.edu.hk}.

38th Conference on Neural Information Processing Systems (NeurIPS 2024).

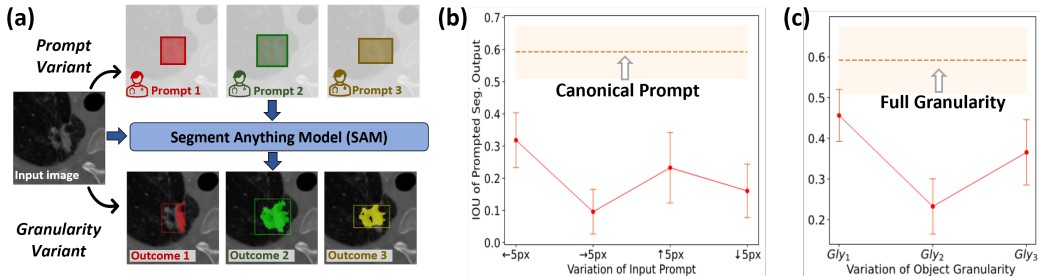

Figure 1: **Analysis of Inherent Ambiguity in SAM**. **(a)**: Feeding SAM with slightly different prompts from multiple experts for a single image can significantly alter the segmentation output. **(b)(c)**: We evaluate SAM using canonical box prompts and various perturbed versions, measuring the mean and variance of segmentation IoU on LIDC. Perturbations involve shifting the box five pixels in different directions and employing various granular outputs within SAM. Results highlight SAM's sensitivity to prompt variations and granularity.

(b). We can observe a significant IoU fluctuation over small prompt differences, which reveals that SAM prediction is highly sensitive to the prompt variants caused by such small perturbation[3].

The second flaw lies in the susceptibility of SAM output to the inherent structural granularity of the object. Despite the fact that VFMs like SAM gain generalizable knowledge and abilities from extensive datasets, they frequently forfeit the capacity to segment specific visual concepts as they are class agnostic, making the model unable to discern the difference between objects with different levels of semantic granularity. Consequently, for targets that are challenging to define, particularly those with rich internal hierarchical granularity, they are inclined to produce multiple candidate results at different granularities and amalgamate them, rather than directly outputting a definitive result. As depicted in Fig. 1 (c), we also discern that the multiple candidate results captured by SAM often exhibit significant differences, and the segmentation precision of outputs at different granularities diverges greatly when compared to the final integrated SAM output that incorporates multiple candidates.

Nonetheless, every coin has two sides. While these observations reveal SAM's faults regarding the output sensitivity, we explore a different perspective: *could this sensitivity flaw become an advantage in other cases, such as ambiguous segmentation, which requires the model to learn from a set of ambiguous object annotations caused by imaging noise, ambiguous tissue boundaries, and different annotator preferences?* Specifically, we are particularly interested in the following: ❶ Given the segmentation result's sensitivity to prompts, can we probabilistically model this prompt variation to tame prompt-sensitive SAM to controllably yield multiple likely results close to the actual fuzzy distribution? ❷ Considering the segmentation result's sensitivity to object structure, can we probabilistically model this granularity variation to tame granularity-sensitive SAM to controllably yield multiple likely results close to the actual ambiguous distribution?

Driven by these questions, we propose an innovative strategy, which flips the inherent category-agnostic ambiguity induced in SAM into a controlled ability to generate a range of feasible results for ambiguous segmentation tasks. Specifically, to simulate segmentation ambiguity under different prompts, we introduce context-aware prompt ambiguity modelling. This method probabilistically models the uncertainty of the prompts inputted into SAM using a latent learnable distribution, which adaptively perceives the specific ambiguity of different contexts. Furthermore, to simulate ambiguity caused by complex object structures at different granularities, we introduce granularity-aware object ambiguity modelling. This method introduces and enhances the visual ambiguity of objects at different granularities into SAM's original image embedding via a learnable embedding distribution. While establishing these two levels of ambiguity modelling, we introduce an efficient optimization strategy based on posterior constraints, allowing the model to mimic models that can perceive the actual ambiguous distribution. Our contributions can be summarized as follows:

---

[3]In practice, such perturbation commonly exists and cannot be controlled by users, even for medical experts.

- We explore the inherent ambiguity in SAM to flip this commonly seen disadvantage in deterministic segmentation tasks into an advantage for more practical ambiguous segmentation tasks that allow multiple possible outputs.

- We propose a $\mathcal{A}$-*SAM* framework that employs a learnable latent distribution to encapsulate the ambiguity at two strata, from prompts and object granularity.

- We introduce an optimization architecture based on variational autoencoders, which effectively represents ambiguity by constraining the sample embedding to align with those from a series of practically feasible annotations.

- Rigurous benchmarked experiments across a wide range of potential scenarios demonstrate that our method produces more accurate, diverse, and reasonable segmentation outputs.

## 2   Related Work

**Prompting Foundational Models for Segmentation.** There has been a surge in the advancement of large-scale vision models for image segmentation, drawing inspiration from language foundation models [62, 4, 25, 39]. Segmentation Foundation Models (SFMs), such as the Segment Anything Model (SAM) [22] and SEEM [66], have delivered significant segmentation results across various downstream datasets [17, 38, 54]. SAM, leveraging a data engine incorporating model-in-the-loop annotation, learns a promotable segmentation framework that generalizes to downstream scenarios in a zero-shot manner. Meanwhile, other models like Painter [56] and SegGPT [57] introduce a robust in-context learning paradigm that enables segmenting any images given an image-mask prompt. On the contrary, SEEM [66] presents a general segmentation model prompted by multi-modal references, such as language and audio, thereby incorporating a wide range of semantic knowledge. These advancements in SFMs, driven by *promptable segmentation* design, involve two types of prompts: semantic prompts (e.g., free-form texts) and spatial prompts (e.g., points or bounding boxes) [22, 57, 40, 41, 43, 26].

Recently, the practice of adapting vision foundation models such as SAM [22] for application in medical image segmentation is garnering increasing interest [46, 10, 12, 44, 33]. A prevalent and cost-effective approach involves adapter techniques, which necessitate the inclusion of bottleneck modules with a finite number of parameters within the model. By fine-tuning these diminutive adapters, SAM can bridge the domain discrepancy between medical and natural images while preserving stellar performance. For instance, models like MSA [58], SAM-Med2D [9], and SAM-adapter [6] utilize adapter strategies to transfigure SAM for medical imaging, thereby achieving significant segmentation outcomes. Despite these advancements, acquiring suitable prompts for SFMs remains largely under-explored. This work aims to explore generating effective prompts for SAM, focusing on harnessing pre-training knowledge to complete ambiguous image segmentation.

**Ambiguous Image Segmentation.** Ambiguous image segmentation aims to model a range of, rather than single, segmentation labels [31, 65, 45]. A wealth of existing research has proposed various techniques to quantify uncertainty. Initial research focused on enhancing a traditional U-Net[51, 16, 29, 5] with a probabilistic component to generate multiple predictions for the same image. This was typically achieved by incorporating a conditional variational autoencoder (cVAE) [53], with the low-dimensional latent space encoding potential segmentation variations. Subsequent work further extended this setup to a hierarchical variant [3, 24, 64, 15]. Other research has utilized normalizing flows to allow for distribution in the cVAE [52, 55] to represent a discrete latent space [49] or incorporated variational dropout and directly used inter-grader variability as a training target. Several other methods [34] do not rely on the Probabilistic U-Net [47, 21, 32, 11, 60]. Monteiro *et al.* [47] proposed a network utilizing a low-rank multivariate normal distribution to model the logit distribution. Kassapis *et al.* [21] leveraged adversarial training to learn potential label maps based on the logits of a trained segmentation network. Zhang *et al.* [63] employed an autoregressive PixelCNN to model the conditional distribution between pixels [37, 36]. Finally, Gao *et al.* [13] used a mixture of stochastic experts, where each expert network estimates a mode of uncertainty, and a gating network predicts the probabilities of an input image being segmented by one of the experts. Unlike previous efforts [1, 42, 59, 7], our approach signifies the first exploration of leveraging the inherent properties in vision foundation models for ambiguous image segmentation.

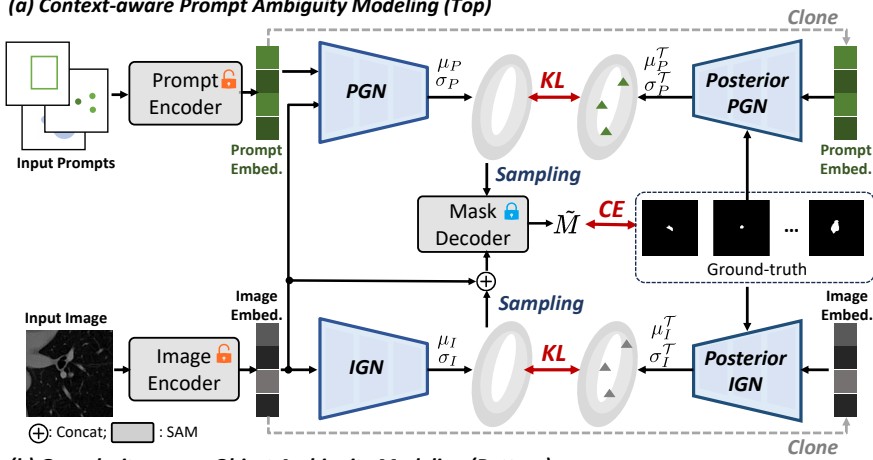

**(a) Context-aware Prompt Ambiguity Modeling (Top)**

**(b) Granularity-aware Object Ambiguity Modeling (Bottom)**

Figure 2: $\mathcal{A}$-*SAM* **Training Pipeline.** We probabilistically model the prompt and object-level ambiguity by jointly probabilities the SAM embeddings with PGN and IGN, respectively.

## 3 Method

### 3.1 Revisiting SAM by Probabilistic Perspective

**SAM: Segment Anything Model.** Given an image $I$ and a set of user-given prompts $P$, which could be a point, a box, or a rough mask, the Segment Anything Model (SAM) [22] employs a vision transformer-based image encoder $\text{Enc}_I$ to extract salient image feature $F_I$ and deploy a prompt encoder $\text{Enc}_P$ with length $k$ to encode prompt embeddings $T_P$, which are denoted as follows,

$$F_I = \text{Enc}_I(I), \quad T_P = \text{Enc}_P(P), \tag{1}$$

where $F_I \in \mathbb{R}^{h \times w \times c}$ and $T_P \in \mathbb{R}^{k \times c}$, where the resolution of the image feature map is represented by $h, w$, and the feature dimension is denoted by $c$. Subsequently, the encoded image and prompts are introduced into the decoder $\text{Dec}_M$ for interaction based on attention mechanisms. SAM constructs the decoder's input tokens by concatenating several learnable mask tokens $T_M$ as prefixes to the prompt tokens $T_P$. These mask tokens are accountable for generating the mask output $M$

$$M = \text{Dec}_M(F_I, T_P, T_M), \tag{2}$$

$\mathcal{A}$-*SAM*: **Lifting SAM to Distributional Space.** Unlike the one-to-one deterministic mapping in SAM, we formulate a probabilistic latent distribution to enable the one-to-many ambiguous mapping named $\mathcal{A}$-*SAM*, with each observation being a sample from this hidden distribution. To this end, the prompt and image embedding can be probabilistically formulated as a distribution:

$$\tilde{T}_P \sim \mathcal{P}_P(\Theta), \quad \tilde{F}_I \sim \mathcal{P}_I(\Phi), \tag{3}$$

where $\mathcal{P}_P$ and $\mathcal{P}_I$ denote a latent distribution for prompt embedding and image embedding, respectively. $\tilde{T}_P$ and $\tilde{F}_I$ denotes a prompt at one sampling from the defined latent distribution at one time. Formally, by implementing multiple rounds of sampling, we can construct a distributional mapping of segmentation outputs with respect to their prompts, formulated as the format of expectation,

$$\tilde{M} = \text{Dec}_M\left(\tilde{F}_I, \tilde{T}_P, T_M\right), \quad st. \ \tilde{T}_P \sim \mathcal{P}_P(\Theta), \tilde{F}_I \sim \mathcal{P}_I(\Phi), \tag{4}$$

where $\tilde{M}$ denotes the SAM output corresponding to one prompt sampling, which can also be interpreted as the sampling from a virtual distribution $\mathcal{P}_M(\Omega)$ for the segmentation results obeying parameters $\Omega$. As a result, we can construct an optimized probability distribution $\tilde{T}_P \sim \mathcal{P}_P(\Theta)$ and $\tilde{F}_I \sim \mathcal{P}_I(\Phi)$ by narrowing the gap between $\tilde{M} \sim \mathcal{P}_M(\Omega)$ and ground-truth distribution.

$\mathcal{A}$-*SAM*: **Inference Stage.** After the training, $\mathcal{A}$-*SAM* can model two types of latent distribution, representing the ambiguity of the prompt variation and the varied object granularity, respectively.

Based on this formulation, each latent sample drawn from the distribution represents a segmentation candidate. Concretely, to predict a set of $m$ segmentations, we apply the network $m$ times to the same input image. In each iteration $i \in \{1, \ldots, m\}$, we draw a random sample regarding prompt embedding $\tilde{T}_P \in \mathbb{R}^{N_P}$ and image embedding $\tilde{F}_I \in \mathbb{R}^{N_I}$ respectively from $\mathcal{P}_P(\Theta)$ and $\mathcal{P}_I(\Phi)$. The final prediction maps $\tilde{M} \sim \mathcal{P}_M(\Omega)$ can be obtained by Eq. 4. In what follows, we will primarily focus on the methodology of training our overall ambiguous segmentation framework.

## 3.2 Context-aware Prompt Ambiguity Modeling

**Distributional Prompt Representation.** To model the distribution of prompt embedding, it is imperative to estimate the parameters $\Theta$ of this distribution. We adopt an axisymmetric Gaussian distribution to characterize the prompt embedding, which is dictated by two crucial parameters, including mean $\mu$ and standard deviation $\sigma$. Then, we can sample a prompt embedding from the given Gaussian distribution, which is shown as

$$\tilde{T}_P \sim \mathcal{P}_P(\Theta) = \mathcal{N}(\mu_P, \mathrm{diag}(\sigma_P)), \tag{5}$$

where $\mu_P$ and $\sigma_P$ denotes the parameters characterized for prompt $P$. $\mu_P$ and $\sigma_P$ respectively denote the mean and standard deviation of the Axis Gaussian Distribution generated by the network, where $\mu_P, \sigma_P \in \mathbb{R}^{N_P}$. This simple yet effective formulation enables the discrete prompt to be continuously represented in the probabilistic latent space, making the uncertainty estimation available.

**Context-aware Prompt Embedding Generation.** To parameterize the latent prompt distribution, we propose a prompt generation network (PGN) to effectively model the aforementioned Gaussian related to prompt embedding. This network is simply designed to include several convolution blocks. Considering the variation in salient regions within the image context, the required prompt positions and sizes should also be varied. Therefore, we incorporate image context as prior knowledge into PGN during the forward inference process. By integrating this prior knowledge, the network can customize a unique prompt-associated axial Gaussian distribution for each image $I$, thereby achieving adaptive and infinite sampling in the latent prompt distribution:

$$[\mu_P, \sigma_P] = F_{PGN}(T_P, F_I; \Theta), \tag{6}$$

where the parameters of the prompt generation network $F_{PGN}$ are modeled by the parameters $\Theta$ of our desired distribution, as each set of generated mean and variance uniquely specifies a distribution. Previous research indicates that allowing the model to conditionally perceive the ground truth label distribution during the training phase enhances training stability for such tasks that exhibit significant uncertainty. Thus, a posterior version for prompt generation network $F_{PGN}^{post}$, parameterized by $\Theta^{\mathcal{T}}$, is further introduced during training, learning to generate the effective distribution for prompt embedding when accessing the ground-truth label distribution:

$$[\mu_P^{\mathcal{T}}, \sigma_P^{\mathcal{T}}] = F_{PGN}^{post}(T_P, F_I, GT; \Theta^{\mathcal{T}}) \tag{7}$$

We only employ this posterior network during training and guide the standard network, which cannot perceive the true labels during testing, to achieve viable performance via a KL loss.

$$\mathcal{L}_{PKL} = D_{\mathrm{KL}}\Big(\mathcal{N}(\mu_P^{\mathcal{T}}, \mathrm{diag}(\sigma_P^{\mathcal{T}})) \,\|\, \mathcal{N}(\mu_P, \mathrm{diag}(\sigma_P))\Big) \tag{8}$$

## 3.3 Granularity-aware Object Ambiguity Modeling

**Distributional Object Representation.** The model integrates object ambiguity from a probabilistic perspective, enhancing its ability to solve problems in innovative ways. We instantiate various segmentation labels to represent ambiguity levels and incorporate them as priors in visual feature extraction, influencing object-related feature modeling for the input image $I$,

$$\tilde{F}_I = \mathrm{Concat}(F_I, \tilde{F}_I\prime), \quad \tilde{F}_I\prime \sim \mathcal{P}_I(\Phi) = \mathcal{N}(\mu_I, \mathrm{diag}(\sigma_I)), \tag{9}$$

where $\mu_I$ and $\sigma_I$ respectively denote the mean and standard deviation of the Axis Gaussian Distribution generated by the network, where $\mu_I, \sigma_I \in \mathbb{R}^{N_I}$. This approach enhances our understanding and depiction of object diversity and uncertainty in a more profound and lucid manner, enabling better adaptability and handling of variation and uncertainty.

**Granularity-aware Image Embedding Generation.** SAM generates multiple candidate segmentation masks for the same object at different granularities and levels, demonstrating the inherent ambiguity of SAM related to object granularity. This inspires us to leverage this aspect to enhance the model's perception of ambiguous objects. We subsequently introduce an image generation network (IGN) for object embedding to model this distribution:

$$[\mu_I, \sigma_I] = F_{IGN}(F_I; \Phi), \tag{10}$$

where the parameters of the image generation network $F_{IGN}$ are modeled by the parameters $\Phi$ of our desired distribution, as each set of generated mean and variance uniquely specifies a distribution. Similar to the previous introduction of a posterior network to enhance learning in modeling prompt ambiguity, we introduce a posterior version of IGN, denoted as $F_{IGN}^{post}$, for perceptible labels.

$$[\mu_I^{\mathcal{T}}, \sigma_I^{\mathcal{T}}] = F_{IGN}^{post}(F_I, GT; \Phi^{\mathcal{T}}) \tag{11}$$

We only employ this posterior network during training and guide the standard network, which cannot perceive the true labels during testing, to achieve viable performance via a KL loss.

$$\mathcal{L}_{IKL} = D_{\mathrm{KL}}\Big(\mathcal{N}(\mu_I^{\mathcal{T}}, \mathrm{diag}(\sigma_I^{\mathcal{T}})) \,\|\, \mathcal{N}(\mu_I, \mathrm{diag}(\sigma_I))\Big) \tag{12}$$

## 3.4 Overall Optimization

In line with current SAM techniques that generate segmentation masks for the same object at different granularity levels, the present approach utilizes this feature to enhance the model's capture of multi-level object concepts through a common ensemble strategy. Specifically, given the multiple candidate outputs from SAM, represented as $\{\tilde{M}^1, \tilde{M}^2..., \tilde{M}^n\}$, where $n$ is the number of scales. By introducing a set of learnable mask weights $\mathcal{W} = \{w_1, w_2..., w_n\} \in \mathbb{R}^n$, the final mask output can be fine-tuned and obtained through a weighted sum calculation:

$$\tilde{M} = \Sigma_{i=1}^n w_i \odot \tilde{M}^i, \tag{13}$$

where $w_1, w_2, ..., w_n$ are initialized to $\frac{1}{n}$ and subsequently fine-tuned to enable the model effectively being aware of the object scales. By adaptively integrating multiple scale masks, the model's perception and modeling capabilities for complex target diversity are further enhanced.

When the representation from SAM is combined with the ground-truth segmentation $GT$ from the training samples, a guide-providing teacher prediction segmentation $\tilde{M}^{\mathcal{T}}$ is created. A cross-entropy loss $CE(\cdot, \cdot)$ is employed to penalize the discrepancies between the distribution of $\tilde{M}^{\mathcal{T}}$ and $GT$, i.e., $\mathcal{P}_M(\Omega)$ and $\mathcal{P}_{GT}$, where the distribution of GT is a constant value that does not need to be parameterised, as: $\mathcal{L}_{Seg} = CE(GT, \tilde{M})$. Additionally, the $KL$ losses introduced to regularize training in prompt ambiguity and image ambiguity, respectively, in Eq. 8 and Eq. 12, are amalgamated into a weighted sum with weight coefficient of $\alpha_P$ and $\alpha_I$:

$$\mathcal{L}_{All} = \mathcal{L}_{Seg} + \alpha_P \cdot \mathcal{L}_{PKL} + \alpha_I \cdot \mathcal{L}_{IKL}. \tag{14}$$

The model is trained from scratch using randomly initialized weights. Parameters requiring update include the prompt encoder and image encoder within SAM as well as PGN, posterior PGN, IGN, and posterior IGN. The KL loss during training aligns the distribution of perceptually true segmentation labels (encoded segmentation variants) with the distribution that is imperceptible at inference time. Adhering to this training objective, the eventual distribution is adjusted to encompass all segmentation variants for a specific input image.

# 4 Experiment

## 4.1 Experimental Setup

**Dataset.** Four datasets are utilized for comparison. The LIDC-IDRI dataset [2] is used for lung lesion segmentation, consisting of lung computed tomography scans from 1010 subjects with annotations from four domain experts. This dataset accurately captures the typical ambiguity found in CT imaging. The BraTS 2017 dataset [18] is used for 3D brain tumor segmentation, comprising 285 cases of 3D MRI images. Each image includes 155 slices and four modes (T1, T1ce, T2, and Flair). These slices

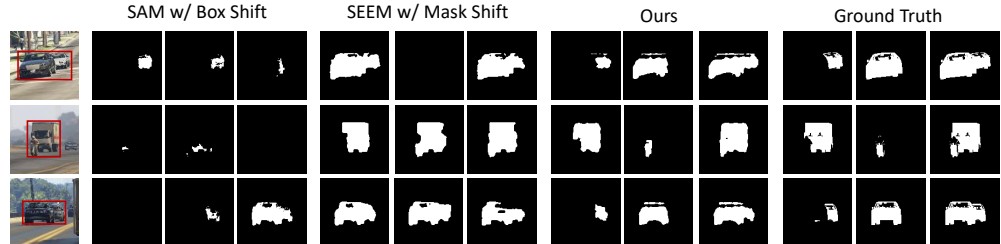

Figure 3: Qualitative comparison with prompted segmentation models adapted for ambiguous segmentation. Examples include three ground-truth expert labels and sampled segmentation masks.

Table 1: Comparison with prompted segmentation models adapted for ambiguous segmentation.

| Metric | GED↓ | HM-IOU↑ | $D_{max}$↑ | $D_{mean}$↑ | GED↓ | HM-IOU↑ | $D_{max}$↑ | $D_{mean}$↑ |
|---|---|---|---|---|---|---|---|---|
| Method | | LIDC | | | | BRATS | | |
| SegGPT w/ Point shift | 0.462 | 0.280 | 0.573 | 0.153 | 0.451 | 0.032 | 0.144 | 0.046 |
| SegGPT w/ Box shift | 0.392 | 0.354 | 0.638 | 0.325 | 0.348 | 0.082 | 0.224 | 0.146 |
| SEEM w/ Mask shift | 0.381 | 0.401 | 0.692 | 0.272 | 0.210 | 0.228 | 0.281 | 0.194 |
| SAM w/ Point shift | 0.377 | 0.365 | 0.650 | 0.337 | 0.252 | 0.169 | 0.334 | 0.238 |
| SAM w/ Box shift | 0.361 | 0.380 | 0.673 | 0.253 | 0.239 | 0.242 | 0.344 | 0.246 |
| $\mathcal{A}$-SAM (Ours) | 0.228 | 0.717 | 0.948 | 0.356 | 0.193 | 0.610 | 0.864 | 0.423 |
| Method | | ISBI | | | | Sim10K | | |
| SegGPT w/ Point shift | 0.649 | 0.662 | 0.659 | 0.323 | 0.259 | 0.128 | 0.151 | 0.127 |
| SegGPT w/ Box shift | 0.527 | 0.772 | 0.874 | 0.536 | 0.272 | 0.152 | 0.220 | 0.183 |
| SEEM w/ Mask shift | 0.522 | 0.821 | 0.908 | 0.760 | 0.238 | 0.271 | 0.344 | 0.246 |
| SAM w/ Point shift | 0.513 | 0.782 | 0.886 | 0.681 | 0.265 | 0.155 | 0.229 | 0.189 |
| SAM w/ Box shift | 0.491 | 0.792 | 0.896 | 0.685 | 0.255 | 0.160 | 0.239 | 0.199 |
| $\mathcal{A}$-SAM (Ours) | 0.276 | 0.835 | 0.926 | 0.904 | 0.233 | 0.637 | 0.851 | 0.327 |

are annotated into four classes: Background, Non-enhancing/Necrotic Tumor Core, Edema, and Enhancing Tumor Core. The ISBI 2016 dataset [14] contains 900 dermoscopic images for training and 379 images for testing, all annotated by an expert with the lesion area. The images are resized and padded to maintain a uniform scale. The SIM 10k dataset [19] consists of 10,000 images rendered by the gaming engine Grand Theft Auto, providing bounding boxes of 58,701 cars in training images.

**Implementation Details.** For the LIDC dataset, we use the included four expert annotations to represent different ambiguous segmentation labels. In the case of the BraTS dataset, we amalgamate annotations from different categories into a binary mask, creating multiple segmentation masks to mimic real-world ambiguous segmentation scenarios. For the ISBI dataset, we use the single label provided. All three datasets are optimized using the Adam optimizer, with a learning rate of 1e-4, over 100 epochs. For the SIM 10k dataset, we select images where pixels from two instances overlap, creating three potential masks. Optimization for this dataset is carried out with Adam optimizer over 500 epochs, with a learning rate of 1e-4. The trade-off coefficients are set as $\alpha_P = \alpha_I = 1$.

**Evaluation Metrics.** Four metrics are used for evaluation: Generalized Energy Distance (GED), Hungarian-Matched Intersection over Union (HM-IoU), Maximum Dice Matching ($D_{max}$), and Average Dice Matching ($D_{mean}$). GED is a metric used in ambiguous image segmentation tasks that compares the distribution of segmentations. It leverages the distance between observations, where lower energy signifies a better match between prediction and the ground truth. HM-IoU calculates the optimal match of Intersection over Union (IoU) between annotations and predictions using Hungarian algorithm, providing an accurate representation of sample fidelity. $D_{max}$ and $D_{mean}$ represent the best and average Dice scores between each prediction result and each ground truth, respectively.

### 4.2 Comparison to Prompted Segmentation Models

Tab. 1 presents the quantitative results on four datasets, offering a comparison with the current state-of-the-art prompting-based segmentation models adapted for ambiguous segmentation tasks. Specifically, we have adapted SegGPT [57], a SAM-like prompt-based segmentation approach that

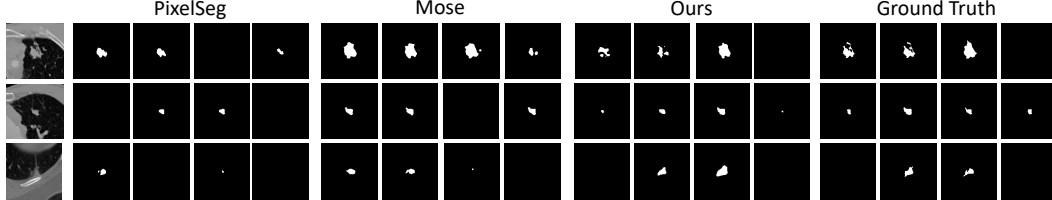

Figure 4: Qualitative comparison with efforts specially designed for ambiguous segmentation. Examples include four ground-truth expert labels and sampled segmentation masks.

Table 2: Quantitative comparison on ambiguous segmentation on the LIDC dataset.

| Method | GED↓ | HM-IOU↑ | $D_{max}$↑ |
|---|---|---|---|
| Prob UNet | 0.324 | 0.423 | - |
| HProb UNet | 0.270 | 0.530 | - |
| PHiseg | 0.262 | 0.595 | - |
| SSN | 0.259 | 0.555 | - |
| CAR | 0.252 | 0.549 | 0.732 |
| PixelSeg | 0.243 | 0.614 | 0.814 |
| CIMD | 0.234 | 0.587 | - |
| Mose | 0.234 | 0.623 | 0.702 |
| $\mathcal{A}$-SAM (Ours) | 0.230 | 0.763 | 0.959 |

Table 3: Quantitative comparison on ambiguous segmentation on the BraTS dataset.

| Method | GED↓ | HM-IOU↑ | $D_{max}$↑ | $D_{mean}$↑ |
|---|---|---|---|---|
| Prob UNet | 0.225 | 0.521 | 0.645 | 0.364 |
| PixelSeg | 0.419 | 0.528 | 0.785 | 0.361 |
| $\mathcal{A}$-SAM (Ours) | 0.192 | 0.603 | 0.886 | 0.438 |

Table 4: Quantitative comparison on ambiguous segmentation on the ISBI dataset.

| Method | GED↓ | HM-IOU↑ | $D_{max}$↑ | $D_{mean}$↑ |
|---|---|---|---|---|
| UNet | - | 0.815 | 0.902 | 0.902 |
| Prob UNet | 0.329 | 0.824 | 0.914 | 0.894 |
| PHiseg | 0.289 | 0.788 | 0.912 | 0.871 |
| cFlow | 0.306 | 0.822 | 0.918 | 0.892 |
| $\mathcal{A}$-SAM (Ours) | 0.267 | 0.834 | 0.918 | 0.905 |

supports multimodal prompt input [30]. To simulate the actual scenario of prompt-based segmentation, we use the smallest box containing all segmentation labels in each image's mask as the standard prompt. The standard prompt is then randomly perturbed multiple times by scaling [0.8,1.2] and shifting up, down, left, or right by [-8,8] pixels to obtain different ambiguous segmentation results. We have also adapted SEEM [66], a segmentation model that follows the in-context learning paradigm. Given a reference image and a reference segmentation mask, it segments the object in the query image. We select an image with multiple mask labels and apply random perturbations on the reference segmentation mask by shifting [-8,8] pixels left, right, up, or down, resulting in multiple different results on the query image. Comparing with SAM, we employ the same paradigm for prompt acquisition and perturbation as SegGPT. We find that $\mathcal{A}$-SAM outperforms the state-of-the-art segmentation foundational models based on point or box in terms of diversity and accuracy. In addition, $\mathcal{A}$-SAM surpasses SEEM, a segmentation paradigm directly based on masks, in all aspects. This indicates that our designed strategy accurately captures the ambiguous attributes present in different images and objects, effectively achieving a balance between diversity and accuracy in ambiguous segmentation tasks. Fig. 3 further illustrates the qualitative results of our method in comparison with existing techniques. Our proposed $\mathcal{A}$-SAM yields segmentations that preserve a greater degree of accurate object detail, particularly boundary specifics, and offers an exceptional visual representation of potential diversity, as compared to other technologies.

### 4.3 Comparison to Conventional Ambiguous Segmentation Models

The numerical outcomes across the four datasets are delineated in Tab. 2, 3, 4, and 5, where we draw comparisons with contemporary leading-edge classical ambiguous segmentation methodologies. Precisely, the comparative method results on the LIDC and ISBI datasets are direct quotations from their respective papers, while outcomes on other datasets are predicated on our reimple-

Table 5: Quantitative comparison on ambiguous segmentation on the SIM 10k dataset.

| Method | GED↓ | HM-IOU↑ | $D_{max}$↑ | $D_{mean}$↑ |
|---|---|---|---|---|
| Prob UNet | 0.292 | 0.391 | 0.462 | 0.229 |
| PixelSeg | 0.398 | 0.525 | 0.644 | 0.358 |
| $\mathcal{A}$-SAM (Ours) | 0.241 | 0.596 | 0.833 | 0.421 |

mentation of their official code. The methods compared encompass recent ambiguous segmentation methodologies that amalgamate a conditional variational autoencoder and UNet: Probabilistic U-Net [23], Hierarchical Probabilistic U-Net (HProb UNet) [24], PhiSeg [3], Stochastic Segmentation Networks (SSN) [47], PixelSeg [63], and the ambiguous segmentation endeavor of Calibrated Adversarial Refinement (CAR) [20] and Collective Intelligence Medical Diffusion (CIMD) [50] that integrate generative models. It also includes ensemble-based techniques utilizing a Mixture-of-expert like the Mix of Stochastic Experts (Mose) [13]. We observe that the $\mathcal{A}$-SAM transcends state-of-the-art methodologies that amalgamate a conditional variational autoencoder and UNet in terms of diversity and precision. Additionally, the $\mathcal{A}$-SAM outstrips segmentation paradigms based on generative models or ensembles in all dimensions. This suggests that our strategy accurately apprehends the ambiguous characteristics inherent in varied images and objects, effectively attaining equilibrium between diversity and precision in the ambiguous segmentation task. Fig. 4 further elucidates the qualitative outcomes of our methodology juxtaposed with extant technologies. In contrast to other technologies, the segmentations engendered by our proposed $\mathcal{A}$-SAM preserve a higher degree of exact object detail, particularly boundary details, and provide a distinctive visual representation of potential diversity.

### 4.4 Further Empirical Results

**Ablation Study.** Tab. 6 delineates the consequences of eliminating various principal strategies of the $\mathcal{A}$-SAM. *No Ambiguity Modeling* precludes all ambiguous modeling, including both prompt embedding and image embedding levels. At this stage, we employ the smallest box encompassing all seg-

Table 6: Ablation study on the proposed key components.

| No Key Components | GED↓ | HM-IOU↑ | $D_{max}$↑ | $D_{mean}$↑ |
|---|---|---|---|---|
| No Ambiguity Modeling | 0.361 | 0.380 | 0.673 | 0.253 |
| No Object Ambiguity | 0.370 | 0.389 | 0.691 | 0.230 |
| No Prompt Ambiguity | 0.308 | 0.674 | 0.930 | 0.336 |
| No Posterior Distillation | 0.266 | 0.385 | 0.805 | 0.341 |
| $\mathcal{A}$-SAM (Ours) | 0.228 | 0.717 | 0.948 | 0.356 |

mentation labels within each image mask as the standard prompt. This standard prompt is then randomly perturbed numerous times by scaling [0.8, 1.2] and shifting up, down, left, or right by [-8, 8] pixels, yielding different ambiguous segmentation outcomes. *No Object Ambiguity* and *No Prompt Ambiguity* either eradicates the ambiguity related to the object or the prompt, that is, it alters its object embedding or prompt embedding to a regular deterministic rather than an ambiguous characteristic. *No Posterior Distillation* eliminates a process of network training guided by a teacher network that can perceive the actual labels. We discover that when any component is excised, the performance correspondingly deteriorates, which underscores the effectiveness of our proposed several strategies.

**Robustness Analysis.** Fig. 5 reports the IoU performance of $\mathcal{A}$-SAM under a variety of instantaneous perturbations. The blue and red solid lines respectively illustrate the performance changes of the SAM model and our method when prompted by different disturbances, while the dashed lines depict the performance of both models under standard prompts, serving as an upper bound. We selected both light and severe degrees of perturbation. Specifically, 'Shift' indicates a random offset of the box by [0,5] pixels, 'Scale' represents a random scaling of the box by [0.85,1.15], 'Shift+' denotes a random offset of the box by [0,8] pixels, and 'Scale+' implies a random scaling of the box by [0.7,1.3]. Compared to the vanilla SAM baseline, $\mathcal{A}$-SAM demonstrates robustness against various instantaneous perturbations.

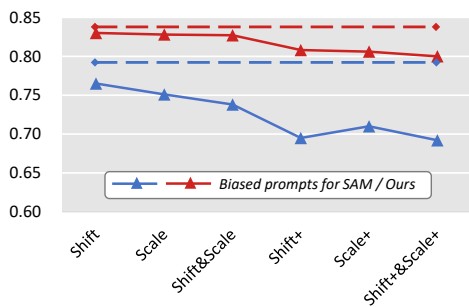

Figure 5: Robustness analysis of our $\mathcal{A}$-SAM framework over the SAM baseline against prompt perturbation.

## 5 Conclusion

The continued evolution of vision foundation models like the Segment Anything Model (SAM) demonstrates impactful universality, while also posing challenges in producing ambiguous and uncertain predictions. Minor changes in the prompt can cause significant variations in the model's output, challenging its required robustness. While many works aim to stabilize SAM's prediction capabilities,

this paper uniquely explores leveraging this perceived flaw to advantageously model inherently ambiguous data distributions. We introduce an innovative optimization framework grounded in a conditional variational autoencoder, which cohesively models the prompt and the object granularity with a latent probability distribution. This approach endows the model with the capacity to adaptively perceive and represent the genuine ambiguous label distribution, thereby enabling SAM to generate a controlled series of diverse, persuasive, and reasonable segmentation outputs. Our comprehensive experiments across multiple practical deployment scenarios involving ambiguity underscore the exceptional performance of our framework, thereby illuminating the need for increased focus on addressing related challenges and opportunities.

**Acknowledgement.** This work was supported by the Hong Kong Research Grants Council (RGC) General Research Fund under Grant 11211221, 14204321. This work was also supported in part by the National Natural Science Foundation of China under Grant 82172033, Grant 82272071 and in part by the Dreams Foundation of Jianghuai Advance Technology Center, and in part by the Open Fund of the National Key Laboratory of Infrared Detection Technologies.

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

# A   Appendix

The following contents are provided in the supplements:

- Limitation and Border Impact.

- More experimental details (Sec. 4.1 in main paper).

- Details of network architecture of $\mathcal{A}$-*SAM*.

- More visualization about our experiments. (Sec. 4.2 and Sec. 4.3 in the main paper).

## A.1   Limitation and Border Impact

**Limitation.** The current approach is constrained by the uncontrolled and unquantifiable nature of uncertainty. This limitation means that the accuracy of handling uncertainty varies across different scenarios. Further systematic analysis is required to comprehend the underlying factors that result in some scenarios being more manageable than others in terms of uncertainty.

**Broader Impacts.** Ambiguous segmentation and uncertainty handling are essential in fields such as image processing and medical diagnostics. Fuzzy segmentation improves our understanding of complex or unclear image content. Proper uncertainty management can enhance prediction accuracy and decision-making, especially in medical diagnostics, aiding in precise disease diagnosis and treatment planning. Ensuring image processing safety is also critical for user privacy and data security, thereby building trust and satisfaction.

## A.2   Detailed Experimental Setup

**Dataset.** Four datasets are used for comparison. **LIDC-IDRI** [2] is a dataset for lung lesion segmentation, which encompasses a voluminous collection of lung computed tomography scans from 1010 distinct subjects, with manual annotations provided by a panel composed of four domain experts. A diversified panel of 12 radiologists leveraged their expertise to provide annotation masks for the dataset, a characteristic that allows the dataset to reflect the typical ambiguity frequently encountered in CT imaging, thereby ensuring comprehensive, accurate annotations that represent a broad range of expert opinions. The resolution of all images is $128{\times}128$. **BraTS 2017** [18] is a dataset for 3D brain tumor segmentation, which consists of 285 cases of 3D MRI images, each image comprising 155 slices. Each slice exhibits four modes (T1, T1ce, T2, and Flair) and is meticulously annotated by professional radiologists into four classes: Background (BG), Non-enhancing/Necrotic Tumor Core (NET), Edema (OD), and Enhancing Tumor Core (ET). The resolution of all images is $240{\times}240$. **ISBI 2016** [14] is a dermoscopy dataset containing 900 dermoscopic images for training and 379 images for testing. Each image is 8-bit RGB and is annotated by an expert with the lesion area. To keep each image at the same scale, we follow [61] to resize the images to $256{\times}192$ and pad the top and bottom with 32 pixels, respectively, to get $256{\times}256$ images. **SIM 10k** [19] consists of 10,000 images that are rendered by the gaming engine Grand Theft Auto. In SIM 10k, bounding boxes of 58,701 cars are provided in the 10,000 training images. All images are used in the training.

**Implementation Details.** For LIDC, we directly employ the four expert annotations included in the dataset to represent four different ambiguous segmentation labels [2]. For BraTS, we overlay and amalgamate annotations from disparate categories, subsequently transforming the outcome into a binary mask that comprises solely the foreground and background [18]. This process is geared towards generating multiple segmentation masks to mimic real-world ambiguous segmentation scenarios, thereby augmenting the rigor and reliability of the experiment. For ISBI, we directly use the single label included in the dataset [14]. For the aforementioned three datasets, we carry out optimization using the Adam optimizer, with a learning rate of 1e-4, over 100 epochs. For the SIM 10k dataset, we contemplate a practical overlap setting. Specifically, we select images in which pixels from two instances within a frame overlap, thereafter creating three potential masks from the two overlapping instances. These masks could represent the first object, the second object, or the union of both. For SIM 10k [19], we carry out optimization using the Adam optimizer with a learning rate of 1e-4, over 500 epochs. The trade-off coefficients in Eq. 14 are set as $\alpha_P = \alpha_I = 1$.

**Evaluation Metrics.** Four metrics are used for strict evaluation. **Generalized Energy Distance (GED)** is a commonly used metric in ambiguous image segmentation tasks that leverages distance between observations by comparing the distribution of segmentations [23], as $D_{GED}^2(P_{gt}, P_{out}) = 2\mathbb{E}[d(S, Y)] - \mathbb{E}[d(S, S')] - \mathbb{E}[d(Y, Y')]$, where $d$ corresponds to the distance measure $d(x, y) = 1 - IoU(x, y)$, $Y$ and $Y'$ are independent samples of $P_{gt}$ and $S$ and $S'$ are sampled from $P_{out}$. Lower energy indicates better agreement between prediction and the ground truth distribution of segmentations. **Hungarian-Matched Intersection over Union (HM-IoU)** is used by calculating the optimal match of Intersection over Union (IoU) between annotations and predictions, which is searched by Hungarian algorithm. This metric offers a more accurate representation of

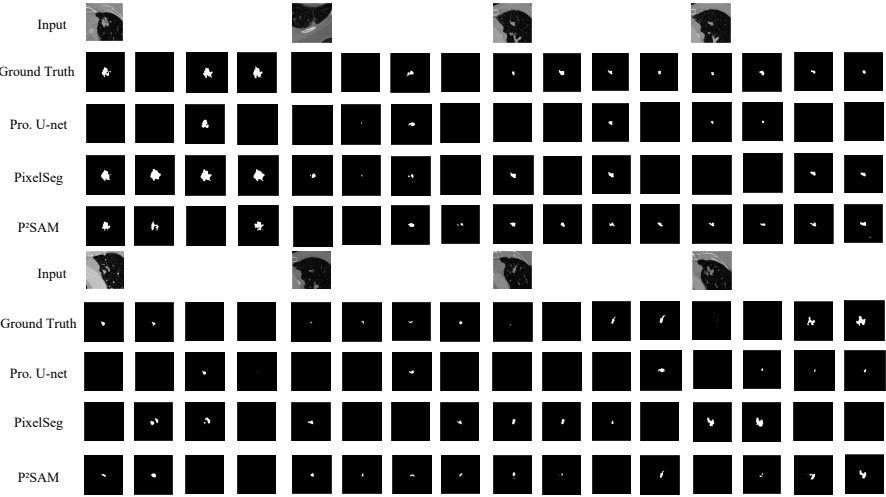

Figure 6: More visualization on the LIDC dataset, displaying only the first 4 samples.

sample fidelity, contrasting the Generalized Energy Distance (GED) which tends to over-reward sample diversity. The Hungarian algorithm identifies the best one-to-one correspondence between objects in two sets. In this context, we utilize $IoU(Y, Y')$ to determine the similarity between two samples. **Maximum & Average Dice Matching ($D_{max}$ & $D_{mean}$)** is respectively the best and average results over the Dice scores between each prediction result and each ground truth.

## A.3 Network Architecture

### A.3.1 Prompted SAM

In the experiment, we adopted the Vit-b version of the SAM model and accommodated $\text{Enc}_I$ by reducing the size of the output feature map $F_I$ by $\frac{1}{8}$ compared with the original. This change is expected to reduce the required memory usage during the training process and accelerate the inference speed of the model. In addition, we adjusted the SAM model to multi output mode with 8 outputs, and set the pixel mean and pixel std parameters to 0 and 1, respectively.

### A.3.2 Prompt Generation Network (PGN)

The network mainly consists of two parts. (1) Encoder: This part contains 4 convolutional blocks, each with 3 convolutional layers inside. These 4 convolutional blocks have channel numbers of 32, 64, 128 and 192, respectively, to gradually extract and deepen features. (2) Axis Gaussian Generation Network: This network consists of a 1x1 convolutional layer with 256 channels and an axial Gaussian distribution generator. This design first increases the dimensionality of the feature map output by the Encoder through a 1x1 convolutional layer to obtain 256 dimensional $\mu$ and $\sigma$, and then these two parameters are fed into a Gaussian generator to generate the distribution of $\tilde{T}_P$.

### A.3.3 Diversity-aware Assembling Module

In our experimental design, we set the number of mask weights $\mathcal{W}$ to 8 and initialize each weight to $\frac{1}{8}$. This setting aims to correspond to 8 outputs of SAM model. In the first stage of the experiment, these weight $\mathcal{W}$ will be trained to meet the model requirements. In the second stage, we will freeze these weights. This is to enable the prompt generation network to generate more diverse and representative segmentation results, thereby effectively guiding the modeling of $\tilde{T}_P$.

## A.4 More Visualization about Experiments

As demonstrated in Fig. 6 and Fig. 7, these illustrations provide an extensive visualization of our research outcomes. These figures meticulously depict various aspects of our data, aiding readers in gaining a profound understanding of our research findings.

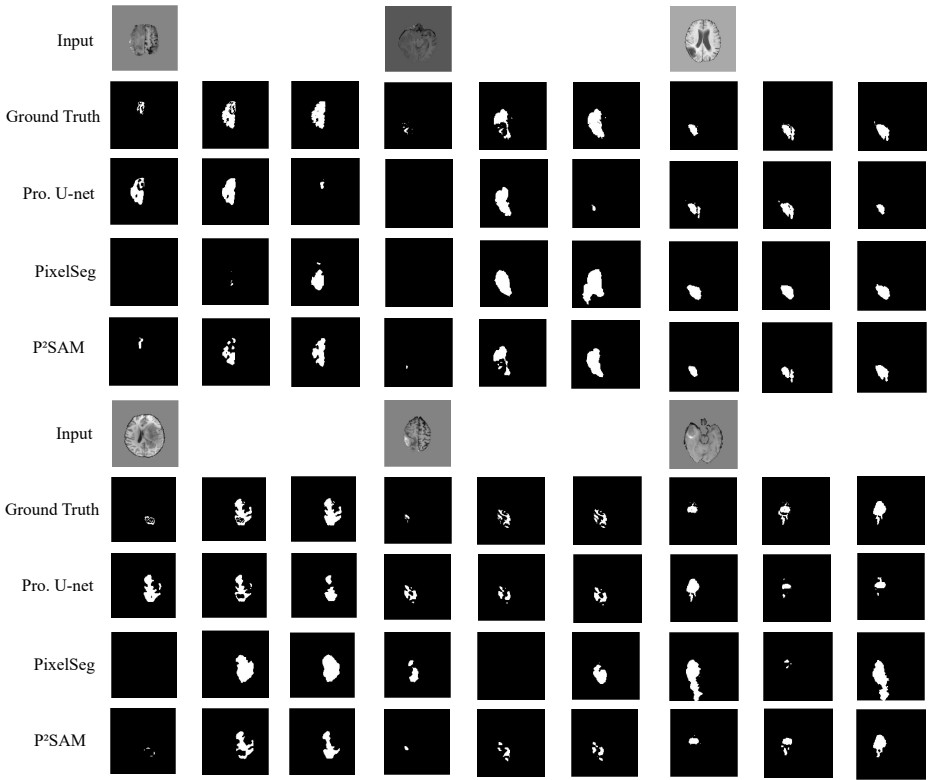

Figure 7: More visualization on the BraTS2017 dataset, displaying only the first 4 samples.

