# OpenReview forum: "Flaws can be Applause: Unleashing Potential of Segmenting Ambiguous Objects in SAM"
_NeurIPS.cc/2024/Conference — NeurIPS 2024 poster_

### Official Review · Reviewer_V2bw · 2024-07-09

**Soundness:** 2
**Presentation:** 3
**Contribution:** 2
**Rating:** 5
**Confidence:** 4

**Summary:**

The paper presents a novel approach to handling the inherent ambiguities in the SAM used for image segmentation. SAM, despite its robustness, often exhibits sensitivity to slight variations in prompts and object granularity, leading to inconsistent predictions. The authors propose a new framework leveraging a conditional variational autoencoder to model these ambiguities probabilistically. This approach enables SAM to produce diverse and reasonable segmentation outputs by adapting to the inherent ambiguities in the data. The paper details extensive experiments demonstrating the effectiveness of this framework across various practical scenarios involving ambiguous segmentations.

**Strengths:**

1.	This work addresses a critical challenge in image segmentation, especially in medical imaging and other fields where ambiguous data is common. By turning SAM's sensitivity into an advantage, the paper contributes to the advancement of robust and adaptable segmentation models.

2.	provides a thorough analysis of SAM's sensitivity to prompt variations and object granularity, backed by detailed experiments and statistical evaluations.

3.	The paper is well-structured, with clear definitions and explanations of the proposed methods. The use of figures and tables enhances the understanding of the framework and its performance.

**Weaknesses:**

1.	The paper primarily tests the framework on specific medical imaging and synthetic datasets. There is a lack of diverse real-world datasets, such as those from different domains (e.g., natural scenes, industrial applications), which might exhibit different types and degrees of ambiguity.

2.	I have a concern that the framework might be overfitted to the specific characteristics of the tested datasets. This concern is evidenced by Table 6, where the "No Prompt Ambiguity" configuration demonstrated metrics comparable to those of A-SAM. Would it be possible that the test datasets might be biased, exhibiting little ambiguity in prompts?

**Questions:**

1.	Equation 13 mentions learning weights to assemble multiple masks into a final output. Where are these weights predicted from? Does the method obtain multiple results through random sampling or a weighted averaging process? If it's the latter, how does it learn multiple sets of weights? If it's random, how does it correspond to the ground truth?

2.	What is the average inference speed for the entire dataset? What percentage of the images contain reasonable masks?

3.	Can you elaborate more on why those specific datasets were being chosen?

4.	Please refer to the weakness section, can you be more specific on what datasets were used in Ablation and Robustness studies?

**Limitations:**

The authors have addressed their limitations and discussed the broader impacts.

---

> ### Author Rebuttal · Authors · 2024-08-05
>
> Thanks for appreciating our paper as addressesing a critical challenge, contributing to the advancement of robust and adaptable segmentation models. We provide pointwise responses to your concerns below.
>
> ## Q1. Applicability to real-world non-synthetic and non-medical datasets
>
>  As shown in Fig. 3 & Tab. 1 in initial submission, we evaluated our method on the Sim10k dataset, featuring synthetic non-medical street scenes with ambiguous semantics. This challenging context demonstrates our approach's broader applicability beyond medical imaging.
>
> Following your suggestion, we further conducted experiments on KINS dataset for instance segmentation in real-world street scenes. More details and results are shown in Gloabl Table 2 and Global Response 2. Our method performs competitively on KINS, still achieving the superior performance. We will include the results in the revision. Thank you for your insightful suggestions!
>
> ## Q2. In Table 6, the performance of "No Prompt Ambiguity" is comparable to the full model?
>
> We respectfully point out that this is a factual error based on the empirical evidence presented in our paper. In Table 6, the results for the "No Prompt Ambiguity" variant are significantly lower across all metrics compared to our full model $\mathcal{A}-SAM (Ours)$, clearly demonstrating the necessity and effectiveness of all components in our proposed design. For example, GED↓: 0.308 vs. 0.228, HM-IOU↑: 0.674 vs. 0.717, etc. These substantial differences consistently observed across various performance indicators provide strong support for the critical role of each element in our model, including the prompt ambiguity mechanism.
>
> ## Q3. Where are the ensembling weights in Eq. 13 obtained?
>
> The original SAM outputs multiple predictions for each prompt to address ambiguity. We manually design an ensemble weight for each prediction and make this weight vector learnable. This approach allows our model to dynamically adjust the importance of each prediction, optimizing the combination of multiple outputs to produce more accurate final segmentations for ambiguous cases.
>
> ## Q4. Inference speed
>
> Thank you for your valuable suggestion! We've added the inference speed for different methods, as shown below. We find that our method achieves better performance while using less or comparable inference time, demonstrating the superiority and practicality of our approach. This balance of high accuracy and computational efficiency highlights the real-world applicability of our $\mathcal{A}$-SAM model.
>
> | Method | GED ↓ | HM-IOU ↑ | D_max ↑ | Avg. Inference Time (s) |
> |:--|--:|--:|--:|--:|
> | Prob UNet | 0.324 | 0.423 | - | 0.0277 |
> | PixelSeg | 0.243 | 0.614 | 0.814 | 3.6093 |
> | Mose | 0.234 | 0.623 | 0.702 | 0.0955 |
> | $\mathcal{A}$-SAM (Ours) | **0.230** | **0.763** | **0.959** | 0.0847 |
>
> ## Q5. What percentage of the images contain reasonable masks?
>
> These diverse ambiguous masks are obtained from expert multi-annotations provided with the established datasets. Currently, there are no existing standards or methods to evaluate their reasonableness. This lack of standardized evaluation metrics for ambiguous annotations is a recognized challenge in the field. Exploring methods to assess the validity and quality of ambiguous masks could indeed be valuable future work. We appreciate your insightful suggestion, as it highlights an important area for further investigation in ambiguous image segmentation.
>
> ## Q6. Why select these datasets?
>
> Following the current common setting established in existing wisedom [1-4], we selected datasets specifically designed for evaluating ambiguous segmentation tasks. This approach ensures our work aligns with the standard practices in the field and allows for meaningful comparisons with existing methods. These datasets are widely recognized in the research community for their ability to challenge models with ambiguous segmentation scenarios.
>
> [1] A probabilistic u-net for segmentation of ambiguous images.
>
> [2] A hierarchical probabilistic u-net for modeling multi-scale ambiguities.
>
> [3] Phiseg: Capturing uncertainty in medical image segmentation.
>
> [4] Stochastic segmentation networks: Modelling spatially correlated aleatoric uncertainty.
>
> ## Q7. What datasets used in Ablation and Robustness studies?
>
> Ablation and Robustness studies are conducted on the LIDC dataset. This dataset was chosen for its comprehensive collection of lung CT scans, each annotated by multiple expert radiologists, providing a rich source of ambiguous segmentations. These studies evaluate the contribution of each model component and assess performance stability under various conditions.

---

> ### Author Response · Authors · 2024-08-10
>
> Dear Reviewer V2bw,
>
> We would greatly appreciate it if you could review our response by Aug 13 AoE. After that date, it might be challenging for us to engage in further discussions. If you have any follow-up questions, please don't hesitate to reach out. We deeply value your expertise and time.
>
> Best,

---

> > ### Comment · Reviewer_V2bw · 2024-08-10
> >
> > Thanks the authors for the responses. They've addressed my concerns. Thus I'll raise my rating.

---

> > > ### Author Response · Authors · 2024-08-14
> > >
> > > Dear Reviewer V2bw,
> > >
> > > We sincerely appreciate your prompt response, valuable suggestions, and rasing the rating for our work. We look forward to including the suggested changes and hope the paper can inspire a broader audience thanks to your constructive feedback!
> > >
> > > Yours,
> > >
> > > Authors

---

### Official Review · Reviewer_2dZP · 2024-07-10

**Soundness:** 3
**Presentation:** 2
**Contribution:** 3
**Rating:** 5
**Confidence:** 3

**Summary:**

This paper proposes a SAM-based framework to address the ambiguous image segmentation problem. The authors present an optimization framework based on a conditional variational autoencoder, which simultaneously models the prompt and the granularity of the object using a latent probability distribution. This approach allows the model to adaptively perceive and represent the real ambiguous label distribution, enabling SAM to controllably produce a series of diverse, convincing, and reasonable segmentation outputs. Experiments on multiple datasets and metrics demonstrate the effectiveness of the method.

**Strengths:**

1. To the best of my knowledge and as indicated by the authors, this paper is the first work that leverages the inherent properties in vision foundation models (SAM) for ambiguous image segmentation.
2. The experimental results demonstrate impressive advantages. Compared to the original SAM, the proposed method shows significantly better performance in the presence of prompt shifts. This high level of robustness is extremely valuable in practical applications.

**Weaknesses:**

1. The task setup of ambiguous image segmentation in this paper is somewhat confusing for me. I have read some referenced and comparative works cited in the paper, such as [a], and found that their task objective is providing multiple segmentation hypotheses for ambiguous images. However, this paper seems to focus more on increasing the accuracy and stability of the model's output when the input prompt has noise or shifts. More explanation about the task setup is needed. Accordingly, it is recommended to include a section in the main text that introduces the task setup, which can help readers who are not experts in this research area understand the paper better.

2. The comparison with conventional ambiguous segmentation models seems unfair because most of the compared methods do not use a network structure as large as SAM. Therefore, it is unclear whether the performance advantage comes from the increased number of network parameters in SAM or from the innovative designs proposed in this paper. I noticed that some of the compared methods, such as [b], can be applied with any encoder-decoder-based segmentation models. Thus, the results of these methods using SAM as the segmentation model should also be reported and compared. This would help evaulate whether the effectiveness of the proposed model is solely due to SAM's larger number of parameters.

3. The writing structure of the paper is somewhat unclear, making it a little difficult to read. For example, the inference method is illustrated in Section 3.1, but the training method is introduced in Section 3.4. It is recommended to create a section titled “Training and Inference,” which contains two subsections that respectively introduce the training and inference methods.

Minor Problem:

1. In Line 169, `Previous research indicates that...' should have corresponding citations added.

[a] A Probabilistic U-Net for Segmentation of Ambiguous Images

[b] MODELING MULTIMODAL ALEATORIC UNCERTAINTY IN SEGMENTATION WITH MIXTURE OF STOCHASTIC EXPERTS

**Questions:**

1. Will the proposed method perform better than the original SAM if there is no point/box/mask shift?

2. Why is the proposed method trained from scratch using randomly initialized weights? Would it be better to finetune from the pre-trained SAM?

**Limitations:**

Please see the weaknesses section.

---

> ### Author Rebuttal · Authors · 2024-08-05
>
> We are very glad and appreciate that you had a positive initial impression. Thanks for appreciating our paper as the first work that leverages the inherent properties in vision foundation models for ambiguous image segmentation, demonstrating impressive advantages and value in practical applications. We provide pointwise responses to your concerns below.
>
> ## Q1. Task setting of ambiguous segmentation
>
> Your assessment of the ambiguous segmentation task setting is entirely correct, namely, providing multiple segmentation hypotheses for ambiguous images.
>
> The key insight of this paper stems from revealing an inherent characteristic of SAM: its non-robust output in response to input prompt perturbations. We innovatively leverage this apparent weakness, transforming it to serve the objective of ambiguous segmentation tasks. Relevant experiments are presented in Figures 3, 4 and Tables 1, 2, 3, 4, 5 in our submission.
>
> Furthermore, we discovered that this design for ambiguous segmentation tasks also incidentally improves the model's robustness to prompt perturbations. Related experiments are shown in Figure 5 of our submission.
>
> ## Q2. Performance comparison with Mose
>
> Thank you for your valuable suggestion! We implemented Mose [1] in conjunction with SAM as you recommended. The results of quantitative comparsion for ambiguous segmentation on the LIDC datasetare presented as below. We also include the qualitative comparison in **Figure 1 in uploaded PDF of Global Response**.
>  We can see that our method still achieves superior performance compared to this approach, demonstrating the advantages of our strategy design beyond the use of SAM.
>
>
> | Method | GED ↓ | HM-IOU ↑ | D_max ↑ |
> |:--|--:|--:|--:|
> | Prob UNet | 0.324 | 0.423 | - |
> | CAR | 0.252 | 0.549 | 0.732 |
> | PixelSeg | 0.243 | 0.614 | 0.814 |
> | CIMD | 0.234 | 0.587 | - |
> | Mose | 0.234 | 0.623 | 0.702 |
> | Mose + SAM | 0.236 | 0.662 | 0.777 |
> | $\mathcal{A}$-SAM (Ours) | **0.230** | **0.763** | **0.959** |
>
>
> [1] MODELING MULTIMODAL ALEATORIC UNCERTAINTY IN SEGMENTATION WITH MIXTURE OF STOCHASTIC EXPERTS
>
> ## Q3. Writing structure
>
> Thank you for your suggestion. We agree with this perspective and will reorganize the paper structure as recommended. We will create a new section titled "Training and Inference" in the revision.
> We will ensure this new section clearly articulates both the training and inference processes, highlighting the connections between them while maintaining the integrity of each part. This change will facilitate readers' understanding of our approach.
> We appreciate your valuable feedback once again, as it will significantly enhance the overall quality and clarity of our paper.
>
> ## Q4. Results of original SAM without point/box/mask shift?
>
> Table 1 (Sec. 4.2) compares SAM variants adapted for ambiguous segmentation. As suggested, we also compared these with the original SAM, as shown in **Global Table 1 in Global Rebuttal**. Results show the original SAM's multi-output capability is insufficient for complex, diverse scenarios, leading to suboptimal performance.
>
> ## Q5. Trained from scratch or finetuned?
>
> We strongly agree with your assertion that fine-tuning a pre-trained SAM would be superior to training from scratch, as it would be impractical to expend enormous resources to build a large model like SAM from the ground up. Therefore, our approach indeed involves fine-tuning a pre-trained SAM.

---

> ### Author Response · Authors · 2024-08-10
>
> Dear Reviewer 2dZP,
>
> We would greatly appreciate it if you could review our response by Aug 13 AoE. After that date, it might be challenging for us to engage in further discussions. If you have any follow-up questions, please don't hesitate to reach out. We deeply value your expertise and time.
>
> Best,

---

> > ### Comment · Reviewer_2dZP · 2024-08-10
> > **Thank you for your response**
> >
> > Thank you for your detailed response. A minor question: Line 213 of main paper indicates that "The model is trained from scratch using randomly initialized weights". So is this a writing mistake?

---

> ### Author Response · Authors · 2024-08-10
>
> Dear Reviewer 2dZP,
>
> Thank you for your comment. Regarding line 213 of the main paper, when we stated "training the model from scratch with randomly initialized weights," we meant to convey that we train all weights outside the SAM component (e.g., PGN, IGN, posterior PGN, posterior IGN, etc.) from scratch, while the weights used for the SAM component (e.g., prompt encoder, image encoder, mask decoder, etc.) are initialized with SAM's original weights. We will clarify this point in the revision. Thank you for bringing this to our attention!
>
> Best,
>
> Authors of Submission 830

---

> > ### Comment · Reviewer_2dZP · 2024-08-12
> >
> > Thanks for the response. I think the method is good, but as also indicated by other reviewers (vXP5, YPmC), the writing of this paper needs further improvement. Therefore, I keep my rating unchanged.

---

> ### Author Response · Authors · 2024-08-13
>
> We appreciate your **acknowledgment on our method** and that this is *the first work leveraging the inherent properties of vision foundation models (SAM) for ambiguous image segmentation**. We will make the following revisions in our updated submission:
> * According to your suggestions, we are commited to **fine-tuning the structure in Method**, by intergrating the related contents of our training and inference pipelines into **an unified sub-section ``Training and Inference''**.
> * Also, we are committed to **modifying the statement in Line 213** by: *"We train all weights from scratch except for the SAM components, including the modules of PGN, IGN, posterior-PGN, and posterior-IGN. For the components including prompt encoder, image encoder, and mask decoder in SAM, we initialize them with SAM's original weights before commencing training."*
>
> We sincerely look forward to**incorporating these suggested changes from you** and hope that this paper will **inspire a broader audience thanks to your constructive feedback**.
>
> Yours,
>
> Authors

---

### Official Review · Reviewer_YPmC · 2024-07-12

**Soundness:** 3
**Presentation:** 2
**Contribution:** 3
**Rating:** 6
**Confidence:** 3

**Summary:**

This paper builds a framework for amigous object segmentation on top of SAM prompted with bounding boxes, which is known to be sensitive to small prompt changes.

The framework is based on a VAE, and the main idea is to jointly model the prompt and the object granularity with a latent probability distribution to gain more control over SAM’s output. In practice, the prompt  embeddings  and image embeddings (controlling granularity) are formulated as a distribution.

The method is evaluated on 3 medical imaging datasets and on a synthetic driving dataset, showing superior performance over the baselines.

**Strengths:**

1. The method is the first to use a promptable large-scale pretrained model like SAM for ambiguous image segmentation
2. The methodology is in general clearly written and easy to follow, figure 2 provides a great overview of the method
3. Extensive evaluation and ablations were performed, showing the method’s superior performance compared to baselines on all of the datasets. (the method is not evaluated on any non-medical real dataset though, see weaknesses)
4. The joint modeling of promts and image embeddings of the proposed method is efficient since the probability sampling is only performed after the SAM encoder and thus the image embedding needs to be computed only once (SAM decoder is lightweight)

**Weaknesses:**

1. The paper contains several unclear statements or missing details, which make the reproducibility of the method difficult.
2. The evaluation is carried on a niche domain (medical) or on synthetic datasets only. It is hard to judge the performance of this method in general real-world setting.

**Questions:**

## General remarks
1. Evaluation on a real-world (not synthetic) non-medical dataset would help to show the generality of the method.
2. It would help readability if it was mentioned that the evaluation metrics are defined in the appendix, also it would help to see the related references in the main paper
3. Is there some intuition/more details on why the granularity is modelled within the image embedding?

## Reproducibility
3. How were the trade-off coefficients tuned?
4. More details on how the three masks from overlapping instances on the SIM10k dataset were obtained should be provided.
5. How was the best checkpoint selected? Was there any hyper-parameter tuning?
6. What does  'achieving significant segmentation outcome‘ mean on line 97? Improvement in segmentation performance over SAM without adapter?

## Fig. 1:
7.  SAM outputs multiple predictions for a prompt, how is this handled in Fig. 1a and 1b?
8.  Medical domain is not in the training domain of SAM so higher uncertainity/instability of prediction is expected, maybe it is not the best example to showcase the behaviour.
9.  What are canonical box prompts from description of Fig. 1? Ground truth bounding boxes?
10.  I assume granularities in 1c correpsond to the three output masks of SAM, what is full granularity then?
11.  The prompt variation experiment depicted in Figure 1 includes bounding boxes that do not cover the whole region to be segmented. It is not unrealistic to control for that in real scenarios, and it would be interesting to see how the figure would change since SAM seems to be quite sensitive to whether the whole object is covered or not – making the bounding box smaller than an object impacts segmentation more than making it larger.
12.  It would be helpful to see how the experts annotate  the example

## Fig 2:
13.  Why is image embedding concatenated with the IGN sample, but the prompt embedding is the output of PGN directly?
14.  Incomplete description – ‚by jointly probabilities‘‘

## Add Weakness 1. – unclear statements and missing details
15. How were the trade-off coefficients set?
16. How exactly was the three masks generated from overlapping instances on the SIM10k dataset?
17. How was the best checkpoint selected? Was any hyper-parameter tuning performed (if yes, on what data)?
18. Line 153 – parameters of axisymmetric Gauss. Distribution „including mean and std“ – the gauss. distirbution does not have any other parameters.
19. What does  'achieving significant segmentation outcome‘ mean on line 97? Improvement in segmentation performance over SAM without adapter?
20. What is meant by the 'final integrated SAM output that integrates multiple candidates‘ on line 44? The only part of SAM that integrates multiple predictions I am aware of is SamAutomaticMaskGenerator class provided by the authors (it features non-maxima suppression) but it prompts SAM with a uniform grid of points while the paper discusses bounding box prompts.
21. The explanation of GT generation for the datasets is confusing since it is incomplete in the paper, it would be nice to at least have a link to the appendix for more details.
22. On lines 38-42, it would help to see an example of such behaviour – what is meant by SAM amalgamating the candidates at different granularities? AFAIK, SAM outputs multiple predictions for each prompt specifically to deal with ambigous prompts.
23. What does 'diminutive adapters‘ on line 93 mean?
24. What is meant by encoder lenght in line 123?

## Add Weakness 2. – evaluation
25. Evaluation on a real-world (not synthetic) non-medical dataset would help to show the generality of the method.
26. Why is original SAM not included in the comparison from subsection 4.2?

## Typos:
27. Line 83 – promotable instead of promptable

**Limitations:**

Limitations are addressed in the appendix.

---

> ### Author Rebuttal · Authors · 2024-08-05
>
> We appreciate your positive initial impression and valuable feedback. We look forward to revising our manuscript based on your suggestions. Below are our point-by-point responses to your concerns. For brevity, we address recurring issues only once.
>
> ## Q1. General remarks
>
> **<Applicable to real-world non-synthetic and non-medical datatset?>**
>  As shown in Fig. 3 & Tab. 1 in initial submission, we evaluated our method on the Sim10k dataset, featuring synthetic non-medical street scenes with ambiguous semantics. This challenging context demonstrates our approach's broader applicability beyond medical imaging.
>
> Following your suggestion, we further conducted experiments on KINS dataset for instance segmentation in real-world street scenes. More details and results are shown in **Gloabl Table 2 and Global Response 2**.
> Our method performs competitively on KINS, achieving the superior performance.
>
> **<More Details about metrics>**
> We appreciate your suggestion. In the revision, we will include the following content in the appendix:
>
> **Generalized Energy Distance (GED)**: A metric for ambiguous image segmentation comparing segmentation distributions:
> \begin{equation}
> D^2_{GED}(P_{gt},P_{out}) = 2\mathbb{E}[d(S,Y)]-\mathbb{E}[d(S,S')]-\mathbb{E}[d(Y,Y')]
> \end{equation}
> where $d(x, y) = 1 - IoU(x, y)$, $Y, Y'$ are samples from $P_{gt}$, and $S, S'$ from $P_{out}$. Lower energy indicates better agreement between prediction and ground truth distributions.
>
> **Maximum and Mean Dice Matching ($D_{max}, D_{mean}$)**: For medical diagnoses, we define Dice score as:
> \begin{equation}
> Dice(\hat{Y},Y) = \begin{cases}
> \frac{2 |Y \cap \hat{Y}|}{|Y|+|\hat{Y}|}, &\text{if } Y \cup \hat{Y} \neq \emptyset\
> 1, & \text{otherwise}
> \end{cases}
> \end{equation}
> We calculate $D_{max}$ for each ground truth $Y_i$ as:
> \begin{equation}
> \{D}_{max} = max \\{Dice(\hat{Y}_1,Y_i),Dice(\hat{Y}_2,Y_i),..., Dice(\hat{Y}_N,Y_i) \\}
> \end{equation}
>
> **Hungarian-Matched Intersection over Union (HM-IoU)**: This metric calculates the optimal 1:1 match between annotation and prediction using the Hungarian algorithm, better representing sample fidelity. It uses $IoU (Y, Y')$ to determine similarity between samples.
>
> **<Insights of modeling granularity within image embedding?>**
> As noted in Lines 185-189, SAM generates multiple candidate masks at different granularities, demonstrating inherent ambiguity related to object granularity. This inspired us to enhance the model's perception of ambiguous objects. We introduce an image generation network (IGN) for object embedding to model this distribution, injecting granularity-aware priors.
>
> ## Q2. Implementation
>
> **<Trade-off coefficients>**
> As noted in Lines 238, the trade-off coefficients are set as α_P = α_I = 1.
>
> **<How three masks from SIM10k dataset were obtained?>**
> As noted in Lines 236-237, for the SIM 10k dataset, we select images where pixels from two instances overlap (e.g., instances A and B), and derive the three potential segmentation masks (e.g., only region A, only region B, the union region of A and B).
>
> **<Model selection>**
> Following current best practices [1], we employ a 6:2:2 ratio for training, validation, and test sets. Results are reported based on best performance on the validation set. We will include this information in our revision.
>
> [1] A probabilistic u-net for segmentation of ambiguous images
>
> ## Q3. Fig1
>
> **<What’s full granularity?>**
> Full granularity refers to the ensemble of various granular outputs within SAM. This approach leverages the multi-scale nature of SAM's internal representations, allowing for a more comprehensive and nuanced segmentation result.
>
> **<Bboxs that are not covering whole region are not considered?>**
> We respectfully disagree, as our approach accounts for this scenario. Due to potential bounding box offsets, the complete segmentation region may not always be fully contained. Our method ensures robust performance even when the initial box doesn't perfectly encapsulate the entire region of interest
>
> ## Q4. More details
>
> **<How GT generated?>**
> We will add the following details to the revised appendix:
>
> The LIDC dataset features manual annotations from four domain experts, accurately reflecting CT imaging ambiguity. Twelve radiologists provided annotation masks. We use the version after the second reading, where experts reviewed and adjusted annotations based on peer feedback, ensuring comprehensive and diverse expert opinions.
>
> The BraTS 2017 dataset includes 285 cases of 3D MRI images (155 slices each) in four modalities (T1, T1ce, T2, Flair). Expert radiologists annotated four classes: background, non-enhanced/necrotic tumor core, oedema, and enhanced tumor core. We overlay these annotations to create binary masks, simulating ambiguous segmentation scenarios.
>
> The ISBI 2016 dataset contains 900 training and 379 testing dermoscopic images. A dermatology expert annotated lesion boundaries in all images, providing a gold standard for segmentation tasks.
>
> For the SIM 10k dataset, we focus on images where pixels from two instances overlap (e.g., instances A and B). From these, we derive three potential segmentation masks: the region exclusive to A, the region exclusive to B, and the union region of A and B.
>
> **<What’s SAM amalgamating different granularities? >**
> We acknowledge that the original SAM outputs multiple predictions per prompt to address ambiguity, without integration. Ensemble integration was introduced in Per-SAM [1]. We will clarify this distinction in our revision.
>
> [1] Personalize segment anything model with one shot.
>
> ## Q5. Evaluation
>
> **<Why original SAM not included in the comparison from Sec. 4.2?>**
> Table 1 (Sec. 4.2) compares SAM variants adapted for ambiguous segmentation. As suggested, we also compared with the original SAM, as shown in **Global Table 1 in Global Response 1**. Results show the original SAM's multi-output capability is insufficient for complex, diverse scenarios, leading to suboptimal performance.

---

> ### Author Response · Authors · 2024-08-10
>
> Dear Reviewer YPmC,
>
> We would greatly appreciate it if you could review our response by Aug 13 AoE. After that date, it might be challenging for us to engage in further discussions. If you have any follow-up questions, please don't hesitate to reach out. We deeply value your expertise and time.
>
> Best,

---

> > ### Comment · Reviewer_YPmC · 2024-08-12
> >
> > Thank you for the response!
> >
> > The following concers remain/arise from your response:
> > * C1: Full granularity refers to the ensemble of various granular outputs within SAM - how is this ensemble computed?
> > * C2: I do not think I stated 'Bboxs that are not covering whole region are not considered' in any part of my review so I am quite confused about this reponse
> > * C3: I do not understand the methodology of creating masks for Sim10k and KINS... .  'the region exclusive to A, the region exclusive to B, and the union region of A and B' - why is the intersection of A, B excluded from the first two masks? I thought the prompt is ambiguous because pixels on the intersection belong to both instances, but here you choose to exclude those pixels... Maybe some examples would help.
> >
> > I think the work is interesting and the method seems to perform well on different datasets. What prevents me from raising my score is that overall, I am worried that the amount of clarifications and rewriting needed is too big.

---

> ### Author Response · Authors · 2024-08-13
> **Further Clarification for Emphasized Issues**
>
> Thank you for your thoughtful comments and the opportunity to clarify our work. We are pleased that our initial response addressed some of your concerns. We will now respectfully address the points you further emphasized.
>
> > C1: Full granularity refers to the ensemble of various granular outputs within SAM - how is this ensemble computed?
>
> Regarding the full granularity ensemble within SAM, as shown in **Eq. (13) and Lines 199-205**, we implement a ensemble strategy to $\textcolor{blue}{\textbf{weighted summing the multi-granular outputs of SAM}}$ to get an ensembled output map. The texts in initial submission is provided as follow: “*Specifically, given multiple candidate outputs from SAM, represented as {M¹, M², ..., M^n}, where n is the number of scales, we introduce a set of learnable mask weights W = {w₁, w₂, ..., w_n} ∈ R^n. The final mask output is obtained through a weighted sum calculation:
> $$
> \tilde{M}=\Sigma_{i=1}^n w_i \odot \tilde{M}^i,
> $$
> where w₁, w₂, ..., w_n are initialized to 1/n and subsequently fine-tuned to enable the model to effectively perceive object scales. By adaptively integrating masks at multiple scales, the model's perception and modeling capabilities for complex target diversity are further enhanced.*”
>
> > C2: I do not think I stated 'Bboxs that are not covering whole region are not considered' in any part of my review so I am quite confused about this reponse
>
> We apologize for the confusion. This response actually **corresponds to Question 11 in your initial review regarding Figure 1**: “The prompt variation experiment depicted in Figure 1…making the bounding box smaller than an object impacts segmentation more than making it larger”. We initially misinterpreted your point, as it requires some efforts to accurately understand. You were inquiring about the **impact of bounding box scaling (both enlargement and reduction) on SAM's segmentation performance, particularly when the box may or may not cover the entire object**, right?
>
> First, in **Table 1 of the initial submission**, both SegGPT and SAM use box shifts that include random pixel offsets in all directions and size scaling from [0.8, 1.2], as stated in **Lines 253-254 and 259-260**. We found that our method $\mathcal{A}-SAM$ can achieve superious results compared with them.
>
> To further specifically address your interest in **how box scaling affects segmentation performance when the box may not fully cover the object**, we conducted additional experiments in LIDC dataset (which follows the same setting in Fig. 1 in initial submission). We scale down and up the canonical box prompt for foreground in each image by different scaling factors from 0.8~1.20.  The results are as follows. We observed that **when the box doesn't fully cover the entire object (i.e., $\textcolor{blue}{\textbf{scaling factor < 1}}$), the performance change is more dramatic compared to the standard box**. This aligns with intuition. We appreciate your insightful suggestion!
>
> | Scaling Factor of Box Prompt | 0.80  | 0.85  | 0.90  | 0.95  | 1.00 (Canonical prompt)   | 1.05  | 1.10  | 1.15  | 1.20  |
> |----------------|-------|-------|-------|-------|-------|-------|-------|-------|-------|
> |Seg. IOU | 0.227 | 0.284 | 0.352 | 0.450 | 0.592 | 0.524 | 0.471 | 0.413 | 0.372
>
> > C3: I do not understand the methodology of creating masks for Sim10k and KINS... . 'the region exclusive to A, the region exclusive to B, and the union region of A and B' - why is the intersection of A, B excluded from the first two masks? …
>
> Regarding the methodology for creating masks for Sim10k and KINS, we refer to region of instance A, region of instance B, and their union A∪B. We didn't extensively discuss the intersection of A and B in the last response, because, for segmentation, there typically aren't strictly overlapping regions at the pixel level. **If a local region of A spatially overlaps and occludes a local region of B, only the local region of A is visible**. Thus, we approach this from a 2D segmentation, pixel classification perspective when referring to 'the region exclusive to A, the region exclusive to B, and the union region of A and B'. From a spatial relationship viewpoint, **$\textcolor{blue}{\textbf{the overlapping region belongs to the occluder}}$ rather than the occluded object**. Thank you for pointing this out!

---

> ### Author Response · Authors · 2024-08-13
> **Further Clarification for Emphasized Issues**
>
> **Last but not least**, we have to acknowledge **the challenge of elucidating all details of the proposed method regarding data, model and metrics** within the **confined space of a submission**. Nevertheless, we beg to point out that the majority of the content addressed in this rebuttal was $\textcolor{blue}{\textbf{already existed in our initial submission}}$, both in the **main text and the appendix**. **For instance**, questions regarding **<more details about metrics>** were addressed in the $\textcolor{blue}{\textbf{appendix, Section A.2, Lines 513-525}}$. The inquiry about **<How multiple GT generated?>** was covered in the $\textcolor{blue}{\textbf{appendix, Section A.2, Lines 485-500}}$. It's worth noting that most of the datasets used in this paer, including LIDC, BraTS, ISIBI, etc., are widely recognized benchmark datasets for ambiguous segmentation, following **established practices [1-5]**. We simply utilized the pre-defined multiple GTs provided by these datasets. Additionally, **<Trade-off coefficients>** were mentioned in $\textcolor{blue}{\textbf{Line 238}}$ in main text. **Our responses to these queries primarily  $\textcolor{blue}{\textbf{emphasize where they are in the original text}}$ rather than introducing new clarification or content.**
>
> Hence, we sincerely believe that the **areas requiring further clarification in the revision are primarily limited to the following several aspects**:
> * Adding a definition & clarification of **full granularity** in **Figure 1's caption and in the introduction**.
> * Including **a sentence in Section 4.1 (Dataset, Line 230)** to tell the readers to **refer to more details in the appendix**.
> * Expanding **Section A.2 in the Appendix (Dataset, after Line 500)** with **a sentence to account for how the ambiguous segmentation masks are made**. We will also add a figure to clarify potential misunderstandings about spatial overlap.
>
> $\textcolor{blue}{\textbf{Most other clarifications are extracted from the existing content in the initial submission and appendix.}}$ Furthermore, we sincerely appreciate $\textcolor{blue}{\textbf{your ``Applause" that this paper is interesting}}$ and we sincerely hope that **some "Flaws" related to clarification issues in this paper can be effectively addressed** thanks to your constructive feedback.
>
> Yours,
>
> Authors
>
> [1] A probabilistic u-net for segmentation of ambiguous images.
>
> [2] Phiseg: Capturing uncertainty in medical image segmentation.
>
> [3] Stochastic segmentation networks: Modelling spatially correlated aleatoric uncertainty
>
> [4] Pixelseg: Pixel-by-pixel stochastic semantic segmentation for ambiguous medical images.
>
> [5] Ambiguous medical image segmentation using diffusion models

---

> > ### Comment · Reviewer_YPmC · 2024-08-14
> >
> > Thank you for the detailed response and additional evaluation - after reading it and after careful consideration, I am raising my score.
> >
> > The authors have shown they are willing to work on improving the clarity of the paper and I think it can benefit the community.
> >
> > I would like to point out that because of the amount of things that were not clear, I had a hard time reading the paper and couldn't have reproduced it. This seems to be a common concern among the reviewers, it is not only about the points raised by me specifically. I trust the authors to significantly improve the writing based on their responses in the rebuttal/during the discussion.
> >
> > Final note re non-medical: Thanks, now it is much clearer. Even though it was established by prior work, I am still not sure why merging two objects that are overlapping in 2D is useful, the task seems better motivated within the medical domain to me.

---

> > > ### Author Response · Authors · 2024-08-14
> > >
> > > Dear Reviewer YPmC,
> > >
> > > We sincerely appreciate your prompt response, valuable suggestions, and raising the rating for our work. We look forward to including the suggested changes and modifications and hope the paper can inspire a broader audience thanks to your constructive feedback!
> > >
> > > Yours,
> > >
> > > Authors

---

### Official Review · Reviewer_vXP5 · 2024-07-13

**Soundness:** 2
**Presentation:** 1
**Contribution:** 2
**Rating:** 5
**Confidence:** 3

**Summary:**

This paper aims to convert the flaws in the vision foundation model (e.g., SAM) into advantages for ambiguous object segmentation. To this end, the authors propose a novel framework that employs latent distribution and an optimization architecture. The authors validated the performance of the proposed methods through comprehensive experiments.

**Strengths:**

Unlike existing approaches that aim to stabilize the sensitivity to ambiguous objects in SAM, this paper suggests leveraging the vulnerability for ambiguous object segmentation. The proposed approach seeks to harness SAM's sensitivity, redeemed as a weakness, to address ambiguous and uncertain predictions.

**Weaknesses:**

1. The explanations are unclear and hard to follow. Specifically, it needs further explanation of how to extract the mean and standard deviation from the convolution blocks and how to utilize the ground truth labels in the posterior version of the prompt generation network.
2. Some symbols are used without explanation (e.g., Θ, Φ, N_i, N_p).
3. Missing reference: Previous research at line 169.
4. Since this paper focuses on clinical scenarios for ambiguous object segmentation, it seems unfair to compare the performance without including existing medical segmentation methods such as OM-Net [1], DC-UNet [2], and CE-Net [3].

[1] https://arxiv.org/pdf/1906.01796v2
[2] https://arxiv.org/pdf/2006.00414v1
[3] https://arxiv.org/pdf/1903.02740v1

**Questions:**

1. What is the difference between PGN and posterior PGN?

**Limitations:**

The authors address limitations of this work and broader impact properly.

---

> ### Author Rebuttal · Authors · 2024-08-04
>
> Thanks for appreciating our paper as harnessing SAM's sensitivity, redeemed as a weakness, to address ambiguous and uncertain predictions. We provide pointwise responses to your concerns below.
>
> ## Q1. Method details
>
> **<How to extract the mean and standard deviation from networks?>**
> As noted in Lines 155-157, the mean and standard deviation of the Axis Gaussian distribution are the two vectors output by the neural network. The input to the network can be the image or prompt embedding, and the network is trained to directly output the mean and standard deviation parameters. This allows for efficient extraction of the distributional statistics from the network outputs, which is crucial for downstream probabilistic modeling tasks.
>
> **<How to utilize the ground truth labels in posterior prompt generation network?>**
> As described in Eq.(7), for each iteration of the training process, the method randomly samples one of the possible ambiguous labels from the label set. This sampled label is then input into the posterior prompt generation network, which is responsible for producing prompts that capture the semantics of the given ambiguous label. By iteratively sampling from the set of ambiguous labels and feeding them to the prompt generation network, the model is able to learn to produce effective prompts that can handle the inherent ambiguity in the training data.
>
> ## Q2. Symbol details
>
> Thank you for your pointing out. The parameters Θ and Φ that you mentioned represent the hyperparameters characterizing two distributions, such as the mean and variance of a Gaussian distribution. $N_i$ and $N_p$ are hyperparameters denoting the dimensions of the vectors. We will provide a more detailed description of these parameters in the revision.
> We appreciate your attention to detail and the opportunity to clarify this point.
>
> ## Q3. Missing reference
>
> Thank you! The authors will add the line "Previous research [1] has described..." at Line 169 to provide more context on how the current approach builds upon prior work in this area, as outlined in the referenced publication [1].
>
> [1] A Probabilistic U-Net for Segmentation of Ambiguous Images
>
> ## Q4. Comparison with more existing models
>
> We appreciate your insightful suggestion! It's worth noting that our method is designed for ambiguous segmentation, while the comparison approach you proposed is designed for conventional deterministic segmentation. To enable a fair comparison, we have adapted our method for deterministic segmentation by ensemble averaging the segmentation results from three sampling iterations into a single output.
>
> We have conducted further comparisons on the overlap dataset and the BraTS 2017 dataset, as mentioned in the references OM-Net [1], DC-UNet [2], and CE-Net [3] you cited. The results of these comparisons are listed as below. As we can see from these results, our method still achieves superior segmentation performance under this setting. We believe this additional evaluation addresses your concern and provides a more comprehensive assessment of our method's performance across different segmentation paradigms.
>
> | Metrics | |Dice↑ |  | |Hausdorff95↓ |  |
> |:----------|:--------:|:------------:|:--------:|:------------:|:--------:|:------:|
> | **Category** | **Core** | **Enh. Core** | **Whole** | **Core** | **Enh. Core** | **Whole** |
> | OM-Net | 0.842 | 0.785 | 0.907 | 7.561 | 3.299 | 4.382 |
> | nnU-Net | 0.819 | 0.776 | 0.903 | 8.642 | 3.163 | 6.767 |
> | Wang et al. | 0.838 | 0.786 | 0.905 | 6.479 | 3.282 | 3.890 |
> | Kamnitsas et al | 0.797 | 0.738 | 0.901 | 6.560 | 4.500 | 4.230 |
> | $\mathcal{A}-SAM$ (Ours) | **0.863** | **0.803** | **0.921** | **6.463** | **3.084** | **3.741** |
>
> [1] https://arxiv.org/pdf/1906.01796v2 [2] https://arxiv.org/pdf/2006.00414v1 [3] [https://arxiv.org/pdf/1903.02740v1](https://arxiv.org/pdf/1903.02740v1)
>
> ## Q5. Difference between PGN and posterior PGN?
>
> As noted in Lines 171-173, a posterior version for the prompt generation network $F^{post}_{PGN}$, parameterized by $\Theta^{\mathcal{T}}$, is further introduced during the training process. This posterior prompt generation network learns to generate the effective distribution for the prompt embedding when **accessing the ground-truth label distribution**. The introduction of this additional network component allows the model to better capture the semantics of the ground-truth labels and generate more targeted prompts, improving the performance of models.

---

> > ### Comment · Reviewer_vXP5 · 2024-08-13
> >
> > Thanks to the authors for the response. They have addressed all my concerns. Thus I will raise my initial rating.

---

> ### Author Response · Authors · 2024-08-10
>
> Dear Reviewer vXP5,
>
> We would greatly appreciate it if you could review our response by Aug 13 AoE. After that date, it might be challenging for us to engage in further discussions. If you have any follow-up questions, please don't hesitate to reach out. We deeply value your expertise and time.
>
> Best,

---

### Author Rebuttal · Authors · 2024-08-05

## Global Response 1. Results of original SAM

As suggested by **Reviewer YPmC** and **Reviewer 2dZP**, we have added the results of the original SAM for comparison. As shown in the figure below, SAM (point) and SAM (box) represent the results of the original SAM obtained using different prompts. It can be seen that, compared to these added original SAM methods, our $\mathcal{A}-SAM$ still exhibits superior results.

| Global Table 1. Comparison with different SAM variants for ambiguous segmentation. |
|:-----------------------------------------------------------------------------|

| Metric | GED↓ | HM-IOU↑ | D_max↑ | D_mean↑ | GED↓ | HM-IOU↑ | D_max↑ | D_mean↑ |
|:--|--:|--:|--:|--:|--:|--:|--:|--:|
| Method | LIDC | | | | BRATS | | | |
| SAM (Point) | 0.385 | 0.347 | 0.664 | 0.335 | 0.274 | 0.167 | 0.341 | 0.225 |
| SAM (Box) | 0.376 | 0.372 | 0.695 | 0.248 | 0.248 | 0.235 | 0.368 | 0.238 |
| SAM w/ Point shift | 0.377 | 0.365 | 0.650 | 0.337 | 0.252 | 0.169 | 0.334 | 0.238 |
| SAM w/ Box shift | 0.361 | 0.380 | 0.673 | 0.253 | 0.239 | 0.242 | 0.344 | 0.246 |
| $\mathcal{A}$-SAM (Ours) | **0.228** | **0.717** | **0.948** | **0.356** | **0.193** |**0.610**| **0.864** | **0.423** |
| Method | ISBI | | | | Sim10K | | | |
| SAM (Point) | 0.538 | 0.793 | 0.891 | 0.709 | 0.294 | 0.162 | 0.249 | 0.197 |
| SAM (Box) | 0.526 | 0.806 | 0.899 | 0.713 | 0.312 | 0.176 | 0.261 | 0.208 |
| SAM w/ Point shift | 0.513 | 0.782 | 0.886 | 0.681 | 0.265 | 0.155 | 0.229 | 0.189 |
| SAM w/ Box shift | 0.491 | 0.792 | 0.896 | 0.685 | 0.255 | 0.160 | 0.239 | 0.199 |
| $\mathcal{A}$-SAM (Ours) | **0.276** | **0.835** | **0.926** | **0.904** | **0.233** | **0.637** | **0.851** | **0.327** |



## Global Response 2. Further results on real-world non-synthetic non-medical dataset.


We appreciate the suggestions from  **Reviewer vXP5** and **Reviewer YPmC** to evaluate our method on a real-world non-synthetic non-medical dataset. To address this, we conducted additional experiments on the KINS dataset [1], which is specifically designed for amodal instance segmentation. KINS is derived from the KITTI dataset [2] and includes instance-level semantic annotations. The dataset comprises 7,474 training images and 7,517 testing images across seven object categories.
We followed the same data processing method as in our initial submission to create ambiguous image-label pairs, selecting images with pixel overlap between two instances and creating three potential masks (instance 1, instance 2, and their union). We then evaluated our approach on KINS using the same metrics. The results are presented in the table below:

| Global Table 2. Comparison Results on A  real-world non-synthetic non-medical dataset, KINS. |
|:-----------------------------------------------------------------------------|

| KINS | GED↓ | HM-IOU↑ | D_max↑ | D_mean↑ |
|----------------------|-------|-------|-------|-------|
| SegGPT w/ Point shift| 0.427 | 0.584 | 0.691 | 0.396 |
| SegGPT w/ Box shift  | 0.385 | 0.640 | 0.758 | 0.472 |
| SEEM w/ Mask shift   | 0.362 | 0.695 | 0.812 | 0.518 |
| SAM w/ Point shift   | 0.340 | 0.612 | 0.735 | 0.445 |
| SAM w/ Box shift     | 0.254 | 0.482 | 0.587 | 0.331 |
| $\mathcal{A}$-SAM (Ours)| **0.237** | **0.633** | **0.839** | **0.445** |


These results demonstrate that our method performs competitively on the KINS dataset, a real-world non-synthetic non-medical dataset. Notably, our method achieves the best performance in terms of GED and D_{max} metrics, while maintaining comparable results with other top-performing methods in HM-IOU and D_{mean} metrics.


[1] Amodal instance segmentation with kins dataset, CVPR, 2019

[2] Vision meets robotics: The kitti dataset, The International Journal of Robotics Research


## Notes on added Figures in the uploaded PDF
As suggested by **Reviewer 2dZP**, we include the visualized comparison between the proposed method  with the Mose+SAM (equipping Mose [3] with SAM backbone) as **Figure 1 in the uploaded PDF**. We can see that the compared to Mose+SAM, the segmentations engendered by our proposed A-SAM preserve a higher degree of exact object detail, particularly boundary details, and provide a distinctive visual representation of potential diversity.

[3] Modeling multimodal aleatoric uncertainty in segmentation with mixture of stochastic expert, in ICLR, 2023

---

### Decision · Program_Chairs · 2024-09-25

**Decision:**

Accept (poster)

**Comment:**

This paper received generally positive feedback, including one weak accept and three borderline accepts. The reviewers commend the authors for addressing the ambiguous image segmentation problem with clear and novel methodologies, as well as for the extensive evaluation and ablation studies. The rebuttal effectively clarified many concerns raised in the initial reviews, providing detailed explanations and solid evidence that convinced the reviewers to increase their ratings. The AC concurs with the reviewers and recommends accepting this paper.

However, as noted by Reviewers vXP5 and YPmC, the revised camera-ready version requires further improvements based on the detailed responses in the rebuttal and the discussion. These enhancements will make the paper more robust. Additionally, the inclusion of more discussion and new results in non-medical tasks is necessary in the camera-ready version to further verify its broader applications. We look forward to seeing this work presented at NeurIPS.